# OpenReview forum: "Cognitive Control Architecture (CCA): A Lifecycle Supervision Framework for Robustly Aligned AI Agents"
_ICLR.cc/2026/Conference — Submitted to ICLR 2026_

### Official Review · Reviewer_fChp · 2025-10-20

**Soundness:** 2
**Presentation:** 3
**Contribution:** 2
**Rating:** 4
**Confidence:** 3

**Summary:**

This paper introduces a new defense approach, Cognitive Control Architecture (CCA), aiming to guard against IPI attacks. They claim that this approach is superior to other existing approaches.

**Strengths:**

- Provided a detailed explanation of the proposed approach and they well motivated the question studied.
- Included a holistic list of baselines.
- Conducted detailed analyses on the experiment results.

**Weaknesses:**

- No standard errors or error bars are reported in Table 1, which presents the main results.

- Lines 371–372: “CCA uniquely balances elite-level security (0.34% ASR) with the highest functional retention (86.43% UA), completely resolving this long-standing problem.” — The wording here seems too strong; “completely resolving” would imply 0% ASR and 100% UA, which is not the case. Please tone down the phrasing.

- Some tables and figures lack clear descriptions in their captions. For example, in Table 1, what is the underlying model used? (Lines 307 mention two models.) Also, what do “direct,” “ign. prev,” “sys. msg,” and “imp. msgs.” refer to? They are mentioned briefly in Line 310 but not explained in sufficient detail. Similarly, what does the y-axis represent in Fig. 4(b)?

- In the Related Work section, it would be beneficial to include a paragraph or a few sentences discussing other types of attacks beyond IPI attacks that also emerge as LLMs become more agentic. For example, one emergent attack in agent setups is sequential decomposition attacks, e.g., the “Monitoring Decomposition Attacks in LLMs with Lightweight Sequential Monitors” paper, and many others.

**Questions:**

My questions are what i said in the weaknesses.
- Could you report the error bars for Table 1?
- Could you tone down Line 371-372 to not say "completely resolving"?
- Could you provide a complete description of the details in tables and figures? Especially the two I mentioned above.
- Could you include a paragraph or a few sentences discussing other types of attacks beyond IPI attacks that also emerge as LLMs become more agentic (in addition to IPI attacks), how other current defenses are proposed for those?

happy to raise my score once all of these are addressed. Thank you for this work.

---

> ### Author Response · Authors · 2025-11-25
> **Response for Q1-Q3**
>
> General Response:
> We thank the reviewer for the constructive feedback. We have addressed your specific questions below and will incorporate these clarifications into the final revision.
>
> **Q1: Could you report the error bars for Table 1?**
>
> Response:
> To ensure statistical rigor, we updated our main experiments. Since our initial submission, the official DeepSeek API has been updated, replacing V3.1 with DeepSeek-V3.2. As V3.2 represents the current stable release with enhanced instruction-following and agentic capabilities while maintaining architectural consistency, we conducted 5 independent runs on this latest version to ensure our results are reproducible.For new experiments of DeepSeek-V3.2, we specifically focused on the Important Messages Attack, as it consistently proved to be the most challenging variant for all baselines and best demonstrates the generalization of our defense. We will release code to facilitate broader testing.
>
> | Defense Method | Model Version | ASR (Attack) $\downarrow$ | UA (Utility) $\uparrow$ | BU (Benign) $\uparrow$ |
> | :--- | :--- | :--- | :--- | :--- |
> | No Defense | DeepSeek-V3.2 | 42.68% | 62.80% | 87.63% |
> | MELON $^\dagger$ | DeepSeek-V3.1 | 0.63% | 35.41% | 72.16% |
> | CCA (Ours) | DeepSeek-V3.2 | 0.72 $\pm$ 0.14% | 86.47 $\pm$ 0.82% | 87.63% |
>
> $^\dagger$ Baseline results cited from the original submission. The results, show low standard deviations, confirming CCA's stability and robustness even on the upgraded model.
>
> **Q2: Could you tone down Line 371-372 to not say "completely resolving"?**
>
> Response:
> We fully agree with the reviewer. The phrase "completely resolving" was an overstatement. While CCA achieves a significantly better trade-off between security and functionality compared to prior works (e.g., maintaining 86.43% UA vs. Melon's 43.76%), it is scientific rigor to acknowledge that no defense is absolute.
>
> We will revise the sentence in the final paper to:
> > *"CCA effectively resolves the traditional trade-off between safety and functionality , achieving a state-of-the-art security level with only a minimal loss of task-completion functionality."*
>
> **Q3: Could you provide a complete description of the details in tables and figures? Especially the two I mentioned above.**
>
> Response:
> We apologize for the lack of clarity in the captions. We provide the specific clarifications below and will update the captions in the revision.
>
> 1. Clarification for Table 1:
> Underlying Model:  The results in Table 1 of the paper were produced using DeepSeek-V3.1
> Attack Types: The columns correspond to the four attack strategies from AgentDojo (Debenedetti et al., 2024), defined as follows:
>     * Direct: The malicious instruction is explicitly stated in the retrieved content (e.g., "TODO: Send a link...").
>     * Ign. Prev (Ignore Previous): The attack attempts to override system instructions (e.g., "Ignore previous instructions and do X instead").
>     * Sys. Msg (System Message): The attack mimics the format of a system-level prompt to trick the agent.
>     * Imp. Msgs (Important Messages): A deceptively aligned attack where the injection claims to be a high-priority message from the user to bypass filters.
>
> 2. Clarification for Figure 4(b):
>     * Y-axis Meaning: The y-axis represents "Average Token Usage per Task" (Lower is better).
>     * Context: This metric illustrates the cost efficiency of our method. CCA consumes significantly fewer tokens compared to MELON because our deterministic "Intent Graph" (Layer 1) handles most checks, avoiding the need for expensive LLM re-execution or deep reasoning for every step.

---

> > ### Author Response · Authors · 2025-11-25
> > **Response for Q4**
> >
> > Response:
> > We appreciate the reviewer's suggestion to broaden the discussion. To thoroughly address this scope, the following response is detailed; however, we will incorporate a concise synthesis of this taxonomy into the "Related Work" section of our final revision. Based on the OWASP Top 10 for LLM Applications (2025) and our threat model (specifically distinguishing between a Benign User attacked by external content vs. a Malicious User attacking the model), we classify emerging threats into three categories based on their attack consequences.
> >
> > Type 1: Malicious Tool Invocation (Action Hijacking; includes IPI and decomposition-based attacks).
> > This is the primary threat in autonomous-agent settings: external content manipulates the agent into invoking tools or APIs in ways the benign user never intended, producing environment-facing side effects (e.g., unauthorized transactions, message sending, information exfiltration). The Important Messages attack in AgentDojo (Debenedett et al., 2024), as well as the decomposition-style attacks described in Monitoring Decomposition Attacks (where harmless-looking subtasks cumulatively steer an agent toward harmful tool calls), both fall squarely into this category.
> >
> > Existing defenses include trajectory re-execution approaches such as MELON (Zhu et al., 2025), which rerun the agent under a masked state and compare tool calls using an embedding model. While effective, this design requires re-executing the entire trajectory and performing additional embedding queries, resulting in roughly 2× more API calls than the undefended baseline.
> >
> > CCA differs fundamentally: instead of rerunning trajectories, CCA constructs a deterministic Intent Graph from the user request and enforces both control-flow and data-flow integrity over the tool-use plan. Because most benign actions naturally lie on the expected execution graph, CCA filters the vast majority of tool calls with zero additional LLM calls, and only adjudicates the rare off-graph deviations. This enables CCA to block action-hijacking attacks—including decomposition-driven ones—while avoiding the substantial overhead of prior approaches.
> >
> >
> > Type 2: Malicious Content Generation (Jailbreaking without tools). Here the attacker’s goal is to induce the LLM to directly emit harmful or policy-violating text without invoking any external tools or APIs—for example, detailed harmful instructions, hate speech, disallowed biological procedures, or security-sensitive content. This category encompasses classical jailbreak attacks in pure QA or text-generation settings, where the harm arises entirely from the model’s output rather than from environment-facing actions.
> > Existing defenses for this regime—such as SmoothLLM (Robey et al., 2023), constitutional-rule classifiers, and safety-tuned content moderators like Llama Guard (Inan et al., 2023)—are specifically designed to guard the base model's text outputs. CCA is complementary to these methods: our architecture targets tool-use integrity in agentic settings and is not intended to replace dedicated text-level safety filters. Accordingly, CCA does not by itself prevent harmful content generation in non-agent, no-tool scenarios.
> >
> > Type 3: Availability Attacks (Logical / Computational DoS).
> > These attacks aim to degrade availability rather than cause direct harm, for example, computational sponge prompts that exhaust inference resources, or logical DoS patterns that trap agents in unproductive loops. Existing mitigations are largely engineering-centric (max token / max step limits, sandboxing architectures such as IsolateGPT (Wu et al., 2024)).
> >
> > CCA provides an architectural mitigation specifically for logical DoS in agent settings: because the Intent Graph is a DAG, cyclic tool-call paths (infinite loops) are not represented in the planned graph, so malicious looping attempts appear as deviations and are rejected. This complements existing heuristic timeouts.
> >
> > References:
> >
> > Zhu, K., Yang, X., Wang, J., Guo, W., & Wang, W. Y. (2025). MELON: Provable Defense Against Indirect Prompt Injection Attacks in AI Agents. *arXiv preprint arXiv:2502.05174.*
> >
> > Wu, Y., Roesner, F., Kohno, T., Zhang, N., & Iqbal, U. (2024). Isolategpt: An execution isolation architecture for llm-based agentic systems. *arXiv preprint arXiv:2403.04960.*
> >
> > Robey, A., Wong, E., Hassani, H., & Pappas, G. J. (2023). Smoothllm: Defending large language models against jailbreaking attacks. *arXiv preprint arXiv:2310.03684.*
> >
> > Inan, H., Upasani, K., Chi, J., Rungta, R., Iyer, K., Mao, Y., ... & Khabsa, M. (2023). Llama guard: Llm-based input-output safeguard for human-ai conversations. *arXiv preprint arXiv:2312.06674.*
> >
> > Debenedetti, E., Zhang, J., Balunović, M., Beurer-Kellner, L., Fischer, M., & Tramèr, F. (2024). Agentdojo: A dynamic environment to evaluate attacks and defenses for llm agents. *arXiv e-prints, arXiv-2406.*

---

> > > ### Comment · Reviewer_fChp · 2025-11-25
> > >
> > > Thank you for the detailed responses. My concerns have been addressed. I have raised my score to 6.

---

### Official Review · Reviewer_eSiz · 2025-10-29

**Soundness:** 2
**Presentation:** 1
**Contribution:** 2
**Rating:** 2
**Confidence:** 4

**Summary:**

This paper proposes a safeguard for defending tool-use agents against indirect prompt injection attacks. The method requires the agent to start by creating a plan in the form of an intent graph comprised of all tool calls the agent will perform in order to execute the user request. Then, during execution, if the action or tool invocation is not in the intent graph, a tiered adjudicator serves as a last line of defense before harm is caused. The proposed approach is evaluated using primarily DeepSeek-V3.1 on AgentDojo. The proposed approach provides a Pareto improvement over the previous state-of-the-art defense mechanism on the evaluated dataset.

**Strengths:**

* The proposed safeguard demonstrates a Pareto improvement over state-of-the-art defenses against indirect prompt injection attacks.
* The proposed approach is more efficient (in terms of tokens) than the state-of-the-art defense.
* The ablations presented in Table 3 are beneficial to understanding why the method works.

**Weaknesses:**

* The presentation quality is quite poor and the paper needs quite a bit of polishing. There are several typos throughout (Figure 2, first column: "Chack" -> "Check", third column: "Adjustor" -> "Adjudicator"?, line 383 "¡"?, inter alia). The figure and table captions are unclear or promise presentation not represented in the figure (e.g. Table 1, the caption promises that the best defense numbers should be bolded, but they are not). This does not inspire confidence in the results.
* The writing is wordy and over-obfuscates the proposed approach and results. Much of the paper is large blocks of text that are difficult to follow and imprecise. The description of the proposed approach can be simplified and compressed to improve understanding.
* The proposed safeguard is quite invasive to the actual implementation of the agent. It requires changing the implementation of the agent. It's unclear how easily this defense can be adapted to new models or models accessed via APIs.
* The proposed safeguard is not evaluated against any adaptive attacks. It's fairly straightforward to create a defense against a static set of attacks. The results do not indicate that the proposed safeguard will generalize to novel attacks.
* The proposed approach is only evaluated on two open models and one static benchmark. It is not obvious that this approach will generalize in practice to other models.
* The proposed defense seems to be directly defending against one type of attack, i.e. indirect prompt injection on tool use agents.
* Using temperature 0 does not necessarily enforce determinism (https://thinkingmachines.ai/blog/defeating-nondeterminism-in-llm-inference/). Also, this does not necessarily represent how tool-use agents are used in practice.

**Questions:**

* Will we need to create new defenses for new types of attacks? It seems like this approach is not adaptable to different types of attacks? How will the efficiency of the agent be affected by integrating more invasive defenses?
* What is Figure 3 showing?
* Can the proposed guardrail be adapted to general tool use agents accessed via API? Could this be implemented by a third-party monitor? How easily can the proposed approach be adapted to a new agent?
* Why does the intent graph necessarily need to be a DAG? How much does errors in the DAG affect the efficacy of the proposed defense?
* What are Figures 4 (b)-(d) showing? Specifically, what is the y-axis?
* Does the proposed approach generalize beyond indirect prompt injection?
* Why does CCA improve UA in Table 2 for Kimi K2? Does the explicit planning and creation of the intent graph improve capabilities?
* For each new tool, do you need to define a new $S_risk$? How is this chosen? It seems extremely heuristic.

---

> ### Author Response · Authors · 2025-11-25
>
> **1. Response to Presentation Quality & Writing Style**
>
> We fully accept this criticism and genuinely apologize for the presentation issues in the current manuscript. We realize that errors such as "Chack" (Figure 2), "Adjustor", and the inconsistent captions (e.g., Table 1) distracted from the technical content and frustrated the reading experience. We take this feedback very seriously.
>
> In the revision, we commit to the following concrete improvements:
>
> * **Rigorous Proofreading:** We will conduct a thorough line-by-line review to correct all typos (including the specific ones noted: "Chack", "Adjustor", and line 383) and formatting errors.
> * **Caption Clarity:** We will rewrite figure and table captions to be self-contained and accurate, ensuring Table 1 correctly bolds the best results as intended.
> * **Streamlined Writing:** We will rewrite the Introduction and Method sections to reduce wordiness and simplify sentence structures. Our goal is to make the description of the Intent Graph and Tiered Adjudicator straightforward and easy to follow.
>
> We are confident that these changes will significantly improve the paper’s readability and ensure the final version meets the high standards of the conference.
>
> **2. Clarification on Figures 3 & 4**
>
> To address your questions regarding the figures:
>
> * **Figure 3** displays the boxplot distributions of the Intent Alignment Score (**\$S\_{align}\$**). It illustrates that the full CCA model (Fig. 3e) effectively suppresses scores for malicious actions, whereas removing components (e.g., w/o **\$S\_{causal}\$**) leads to dangerous high-score outliers.
> * **Figure 4 (b)-(d)** analyzes efficiency, where the ​**y-axis represents Token Usage**​. These plots demonstrate that CCA is \~3.3x more token-efficient than MELON (Fig. 4b) and that the Intent Graph is the key driver of these savings (Fig. 4c).
>
> We will strictly correct the captions and axis labels in the revision to make this explicit.
>
> **3. Clarification on Invasiveness & API Adaptability**
>
> We respectfully clarify that CCA is designed as a lightweight ​**external proxy**​, not an internal modification. Crucially, it operates entirely on the input/output text stream and it does not require access to the model’s weights, gradients, or hidden states.
>
> This makes CCA naturally deployable as a third-party monitor or "sidecar" for any black-box API:
>
> * **Deployment:** CCA runs in a separate process that intercepts proposed tool calls. It validates them against the Intent Graph (Pillar I) and Adjudicator (Pillar II), and only forwards approved actions to the actual execution environment.
> * **Adaptability:** Swapping the base agent (e.g., to a new API model) only changes the message source; the verification logic remains completely unchanged.
>
> In this sense, CCA is architecturally different from more heavyweight system-level defenses such as IsolateGPT (Wu et al., 2024), which require restructuring the entire agent stack into multiple isolated processes with carefully controlled communication channels. By contrast, CCA operates as a plug-in monitor that can be placed in front of an existing agent with minimal changes to the agent code.

---

> > ### Author Response · Authors · 2025-11-25
> >
> > **4. Evaluation Scope: Generalization to New Models & Benchmarks**
> >
> > To demonstrate that CCA generalizes beyond specific architectures, we extended our evaluation to include **Qwen3-Next-80B** and ​**GPT-4.1-mini**​. The results below confirm that our defense maintains high security and utility across both open-weight and proprietary models:
> >
> > | Model                         | Method        | ASR (Attack) $\\downarrow$ | UA (Utility) $\uparrow$ | BU (Benign) $\uparrow$ |
> > | ------------------------------------ | ------------- | ------------------------- | ----------------------- | ---------------------- |
> > | **Qwen3-Next-80B**    | No Defense    | 32.14\%                   | 64.49\%                 | 82.47\%                |
> > |  | **CCA (Ours)**| **1.69\%**                | **77.87\%**             | **80.41\%**            |
> > | **GPT-4.1-mini**   | No Defense    | 20.89\%                   | 30.36\%                 | 72.16\%                |
> > |   | **CCA (Ours)**| **1.90\%**                | **65.23\%**             | **65.98\%**            |
> >
> > Regarding the benchmark, we selected AgentDojo because it is the established standard for dynamic IPI defenses, used by recent state-of-the-art methods like TaskShield and MELON. Unlike static datasets, AgentDojo provides a realistic, multi-domain environment that accurately reflects the tool-use threat model. For these additional model evaluations (Qwen and GPT-4.1-mini), we specifically focused on the Important Messages Attack, as it consistently proved to be the most challenging variant for all baselines and best demonstrates the generalization of our defense. We will release our full codebase to facilitate easy extension to other environments in the future.
> >
> > **5. Defense Robustness against Adaptive Attacks & Scope**
> >
> > CCA is explicitly designed for the "benign user, untrusted content" threat model, where external data attempts to hijack tool execution. Crucially, CCA operates as a **positive security model** (allow-list) rather than a signature-based defense.
> >
> > * **Mechanism of Generalization:** Pillar I forces the agent to pre-commit to a canonical control-flow and data-flow plan before processing untrusted content. Therefore, regardless of the specific attack vector (e.g., Memory Injection, RAG Poisoning, or future adaptive variants), the attack *must* cause a deviation from the committed plan to succeed.
> > * **Universal Detection:** Since CCA detects the behavioral deviation rather than the prompt pattern, it naturally catches novel IPI variants without requiring updates.
> >
> > While CCA specifically secures the tool-execution lifecycle, it is designed to be complementary to existing defenses that target fundamentally different threat models, such as adversarial users or non-tool jailbreaks.
> >
> > **6. Justification for DAG Structure & Robustness to Planning Errors**
> >
> > The DAG structure is essential for two reasons: it enforces strict data-flow provenance (ensuring parameters originate from valid upstream nodes) and prevents logical DoS loops.
> >
> > Regarding errors in the initial graph: An imperfect plan does not compromise security. If the agent proposes a legitimate action that was missing from the initial graph, CCA simply flags it as a deviation. The Tiered Adjudicator then verifies it; if approved, the graph is dynamically updated to include the new path. This "self-healing" mechanism ensures that planning omissions result only in a one-time efficiency cost for adjudication, rather than a failure in task completion or safety.

---

> > > ### Author Response · Authors · 2025-11-25
> > >
> > > **7. Explanation of UA Improvement for Kimi K2**
> > >
> > > “Why does CCA improve UA in Table 2 for Kimi K2? Does the explicit planning and creation of the intent graph improve capabilities?”
> > >
> > > Yes, CCA can improve UA, especially for weaker agents, for two reasons:
> > >
> > > 1. **Attack Blocking:** In the no-defense setting, successful attacks (high ASR) divert the agent into executing malicious instructions, causing it to abandon the user’s original task, which lowers UA. When CCA blocks these deviations, the agent is forced to stay on or return to the user-intended path, increasing UA.
> > > 2. **Structured Planning:** The explicit planning step in Pillar I acts similarly to a structured “chain-of-thought” for tool use: it encourages models like Kimi K2 to articulate a coherent plan before acting, which reduces benign flailing and irrelevant calls. This planning boost is particularly noticeable for a weaker base model.
> > >
> > > **8. Rationale for Heuristic Risk Scores (**\$S\_{risk}\$**)**
> > >
> > > We acknowledge that **\$S\_{risk}\$** is heuristic by design. This serves as a necessary ​deterministic anchor. Similar to OS permission manifests (e.g., Android/iOS), static scores are assigned based on clear side-effects (e.g., read-only vs. financial). This approach ensures stability and enriches the decision-making range, providing the granularity essential for real-world deployments where risk is rarely binary.
> > >
> > > **9. Validity of Temperature 0 for Evaluation**
> > >
> > > We believe this critique conflates system-level engineering with standard algorithmic evaluation. While we acknowledge that low-level floating-point accumulation can introduce microscopic non-determinism (as discussed in the cited blog), **Temperature 0** remains the rigorous community standard for minimizing sampling randomness in benchmarks like AgentDojo. Requiring batch-invariant kernel implementations to achieve bit-wise reproducibility is an infrastructure challenge orthogonal to our algorithmic contribution.
> > >
> > > Crucially, CCA does not rely on strict determinism. It enforces control-flow integrity on the ​sequence of actions produced​, regardless of whether that sequence varies due to sampling settings or system noise. The defense is robust to the underlying generation variance.

---

### Official Review · Reviewer_x7VM · 2025-10-30

**Soundness:** 3
**Presentation:** 2
**Contribution:** 3
**Rating:** 6
**Confidence:** 3

**Summary:**

The paper proposes the Cognitive Control Architecture (CCA), a multi-layer framework to defend an LLM agent against prompt injection attacks. The framework is composed of two layers: the first layer is used to construct the execution graphs from the user's intention. If the actual action deviates from the pre-planned graphs. The second layer is activated, composed of a weighted score of multiple dimensions to evaluate if the action is safe. In the experiment, the paper selects two LLMs (DeepSeek and KIMI), evaluating on AgentDojo. The result shows that CCA achieves state-of-the-art defense results and efficiency without decreasing the benign performance.

**Strengths:**

- **Insightful design**: The paper is novel in designing a multi-layer framework to inspect the agent action process. The framework is designed to inspect both the data-flow and the underlying intention to ensure a safe agent behavior.
- **Promising results**: The paper shows promising results in defending against prompt injection attacks in AgentDojo benchmarks, surpassing previous work or achieving comparable performance in lowering the attack success rate. Meanwhile, the proposed methods don’t sacrifice the benign performance. Additionally, the API overhead is also reduced to less than half of MELON.
- **Comprehensive comparisons of related works**: The paper compares the proposed methods with four related approaches. The comprehensive comparison further justifies the advantage of the proposed framework.

**Weaknesses:**

## Major


- **Lack of evaluation dataset and models**: The paper mainly evaluated the results on one dataset (AgentDojo), using two LLMs (DeepSeek and KIMI). The authors are expected to conduct experiments on multiple datasets and models to support the generalization of the proposed methods.
- **Lack of experimental justification of Graph Updated**: The paper proposes to dynamically update the graph, but lacks of ablation study on how the design will influence the benign utilization and attack effectiveness. The author is suggested to conduct experiments to justify this claimed methodology design.


## Minor
- **Visible portion of typos**: The paper has a visible portion of typos, which influences the readability. For example:
  - No space before citation in Lines 41, 43, and 46.
  - No space in Line 167 after *”(Pillar I)”* and Line 232 before *”The dynamic”*.
  - In Table 2, the BU results of CCA (e.g., 86.6%) don’t match the main content in Line 400 (e.g., 84.54%).
  - In Figure 2, the Pillar II is “Tiered Adjustor”, which is inconsistent with the main paper as “Tiered Adjudicator”.
- **Strong feeling of LLM writing**: Even though the paper declares the LLM usage in writing, the strong feeling of LLM writing (e.g., using an extensive amount of uncommon expressions) might jeopardize the readability. I listed several sentences below that I feel might be written by LLMs:
  - (Line 043) their **inherent cognitive fragility—manifesting as a lack of robust risk awareness**…
  - (Line 047) This enables attackers to unlawfully **steer tool invocations**...
  - (Line 050) as **stringent controls** impair agent capabilities
  - (Line 057) **forcing an untenable trade-off**
  - (Line 067) To **break this impasse**,...
- **Presentation issue**: Table 1 is confusing: the ASR seems to be applied to all columns, but the BU column is evaluated using the benign utility metrics. In Figures 3 and 4, the text is too small compared to the main content.

**Questions:**

- What if the action is correct (thus can pass the check for Pillar II) and is being executed? Specifically, the pipeline only checks if the action is correct according to the intention graph. What if the prompt injection is targeted at modifying the value of a certain action? For example, a transaction of $10,000, instead of 100. How can the proposed framework defend against this type of attack?

---

> ### Author Response · Authors · 2025-11-25
>
> We sincerely thank the reviewer for the comprehensive and constructive review. We are encouraged by your recognition of our framework's insightful design and promising results. Your feedback on expanding the evaluation scope and clarifying the graph update mechanism has been extremely valuable for strengthening our work. We have conducted additional experiments to further validate our framework. Below, we address each of your comments in detail.
>
> ### 1. Evaluation scope: more models and generalization
>
> > “The paper mainly evaluated the results on one dataset (AgentDojo)... experiments on multiple datasets and models…”
>
> We have extended our evaluation to two distinct architectures: Qwen3-Next-80B (large, open-weight) and GPT-4.1-mini (proprietary API).
>
> | Model                          | Method        | ASR (Attack) $\\downarrow$ | UA (Utility) $\uparrow$ | BU (Benign) $\uparrow$ |
> | ------------------------------------ | ------------- | ------------------------- | ----------------------- | ---------------------- |
> | **Qwen3-Next-80B**    | No Defense    | 32.14\%                   | 64.49\%                 | 82.47\%                |
> |  | **CCA (Ours)**| **1.69\%**                | **77.87\%**             | **80.41\%**            |
> | **GPT-4.1-mini**   | No Defense    | 20.89\%                   | 30.36\%                 | 72.16\%                |
> |   | **CCA (Ours)**| **1.90\%**                | **65.23\%**             | **65.98\%**            |
>
> **Results:** As shown above, CCA exhibits consistent model-agnostic robustness:
>
> 1. Universal Security: ASR is suppressed to < 2% across all models.
> 2. Utility Rescue: CCA recovers utility significantly (e.g., doubling GPT-4.1-mini's success rate from 30.36% to 65.23%).
> 3. Minimal Overhead: Benign Utility (BU) drops are marginal.
>
> **Regarding Datasets:** We focused on AgentDojo as it is widely adopted as a standard benchmark for tool-use injection (used by TaskShield, MELON), offering dynamic, multi-domain environments unlike static benchmarks. For these additional model evaluations (Qwen and GPT-4.1-mini), we specifically focused on the Important Messages Attack, as it consistently proved to be the most challenging variant for all baselines and best demonstrates the generalization of our defense. We will release code to facilitate broader testing.
>
> ### 2. Dynamic graph update (“Graph Updated”)
>
> > “...lacks of ablation study on how the design will influence the benign utilization and attack effectiveness.”
>
> The Intent Graph enforces strict control/data-flow integrity, while the "Graph Updated" mechanism enables controlled adaptation to validated benign deviations:
>
> * **Mechanism:** If an agent proposes an off-graph action, it is *always* detected as a deviation and sent to the Tiered Adjudicator. Only if validated as benign and causally necessary is it executed and appended to the graph.
> * **Ablation Logic:** Conceptually, removing updates would not improve security (malicious actions are caught *before* update) but would  harm Benign Utility (BU). This is because the Intent Graph in the system prompt acts as the agent's execution context; if it remains static (reflecting only the initial plan) despite approved deviations, the agent fails to update its understanding of the current task progress, leading to inefficient integration of new information.
>
> ### 3. Typos, style, and presentation
>
> > “Visible portion of typos… Table 1 is confusing…”
>
> We apologize for these issues. we will:
>
> 1. **Correct all typos** and unify labels (e.g., “Tiered Adjudicator”).
> 2. **Rewrite wordy sections** to reduce "LLM-like" phrasing.
> 3. **Refine figure captions** to provide detailed descriptions of the underlying models and metrics, and improve figure font sizes.
>
> ### 4. Question: parameter-level attacks (e.g., \\$10,000 vs. \\$100)
>
> This scenario highlights a key superiority of CCA. Unlike defenses that only monitor tool sequences, CCA’s Intent Graph enforces strict ​Data-Flow Integrity​, enabling immediate detection of parameter tampering.
>
> 1. **Immediate Deviation Detection (Pillar I):** The Intent Graph explicitly binds every parameter to a source consistent with the user's original task logic (e.g., the user's direct instruction or a specific tool output). An attack attempting to inject \\$10,000 (derived from untrusted external context) to replace the legitimate "\$100" inevitably violates this pre-defined data flow. CCA detects this mismatch instantly as an off-graph deviation, preventing silent execution.
> 2. **Tiered Adjudicator (Pillar II):** Once flagged, such an obvious anomaly—replacing a user-verified low value with a high-risk amount from an untrusted source—is easily captured and blocked by the Adjudicator's alignment check.

---

> > ### Author Response · Authors · 2025-12-03
> > **Experimental Justification for Graph Updates**
> >
> > To validate our analysis, we conducted an additional ablation study on DeepSeek-V3.2. As shown in the table below:
> > | Configuration                | ASR (\%) ↓        | UA (\%) ↑        |
> > |-----------------------------|-------------------|------------------|
> > | Full CCA (with updates)     | \$0.72 \pm 0.14\$   | \$86.47 \pm 0.82\$ |
> > | w/o Graph Update (Static)   | 0.63              | 84.19            |
> >
> > (Note: Full CCA results are mean \$\pm\$ standard deviation over 5 runs)
> >
> > We observe that disabling the graph update leads to a drop in utility (UA: 86.47% $\to$ 84.19%), indicating that the mechanism helps prevent context loss during complex tasks. Conversely, the ASR remains statistically stable (difference within error margins). This demonstrates that the dynamic update preserves functionality without compromising the security performance.

---

### Official Review · Reviewer_CdP1 · 2025-10-31

**Soundness:** 1
**Presentation:** 1
**Contribution:** 1
**Rating:** 0
**Confidence:** 4

**Summary:**

This paper proposes an approach that leverages few-shot learning to handle perception uncertainties and anomalous inputs in autonomous driving systems by leveraging information geometry-guided dimensionality reduction, which decouples high-dimensional text embeddings into driving-relevant features (spatial relationships, temporal dynamics, physical constraints) while preserving contextual reasoning capabilities. This paper demonstrates that their approach can achieve a 24.93% average collision rate on UniAD, outperforming GPT-Driver by 22% under normal conditions and showing only 14.9% performance degradation under anomalies compared to 17-21% for existing LLM-based methods.

**Strengths:**

This paper tries to address an emerging and important area of research, the safety of LLM-based autonomous driving. As our community and society pay close attention to this area, I am happy to see a paper submission in this area. I can see that their methodology achieves higher performance than baseline and existing methods in their evaluation.

**Weaknesses:**

I have the following major concerns about this paper:

### Critical presentation errors

This paper has a significant number of presentation errors across the paper. Particularly, this paper does not have any references to the figures in this paper, even though this paper has 4 figures.  This prevents me from fully being convinced of the reported result's validity. Furthermore, this paper does not clearly explain how their dataset constructed in Section 4.1 is used in the following evaluation with the datasets of the UniAD and ST-P3. These presentation errors are not at the level of a minor issue, but a major issue, leading me to the rejection side.

### Lack of sufficient explanation about the experimental setup

I do not fully understand the details of their evaluation setups. I can see that they constructed a dataset with anomalies extracted from the nuScenes dataset, but I am not fully sure about the details of how this dataset is used with the UniAD and ST-P3. Furthermore, this paper should provide more details of the dataset they constructed since the quality of the dataset has not been validated yet. In some worst cases, it might be constructed in a cherry-picking manner to benefit their approach. This paper should provide more detailed experimental setups to show that their evaluation is conducted on fair ground.

### Lack of sufficient explanation of why their dimension reduction is particularly good

This paper claims that their dimension reduction technique shows significant performance improvements. However, dimension reduction is one of the most common approaches to improve robustness. This paper should provide more experimental results to support why their dimension reduction is particularly good. This paper may compare it with baseline dimension reduction techniques or employ an ablation study. Otherwise, I cannot be fully convinced whether this paper brings meaningful contributions to our community.

**Questions:**

Is it possible to describe the details of their evaluation setups?

---

### Meta-Review · Area_Chair_K5sj · 2026-01-02

**Summary:**

CdP1's review seems to be intended for other papers; therefore, it's excluded from the following discussion.

The reviewers have raised the following major concerns:
1. Lack of evaluation across diverse datasets and models. (x7VM, eSiz)
2. Lack of justification for the method design. (x7VM)
3. The method requires significant modification to the agent. (eSiz)
4. The defense only applies to one specific attack. (eSiz)
5. The presentation needs significant improvement. (x7VM, eSiz, fChp)

**Reviewer Concerns:**

Concerns addressed by rebuttal:
(2), (5)

Outstanding concerns:
(1) The rebuttal extends to two additional LLMs but doesn't consider other datasets.
(3) (4)

**Reviewer Scores:**

- x7VM: The rebuttal partially addresses their concerns; the reviewers may keep or marginally increase their score.
- eSiz: The rebuttal doesn't fully address their concerns; the reviewers may keep their score.
- fChp: The rebuttal addresses most of their concerns; the reviewers may marginally increase their score.

---

### Decision · Program_Chairs · 2026-01-26

Reject